# Stem Cells Derived from Human Exfoliated Deciduous Teeth Functional Assessment: Exploring the Changes of Free Fatty Acids Composition during Cultivation

**DOI:** 10.3390/ijms242417249

**Published:** 2023-12-08

**Authors:** Alexandra Ivan, Mirabela I. Cristea, Ada Telea, Camelia Oprean, Atena Galuscan, Calin A. Tatu, Virgil Paunescu

**Affiliations:** 1Department of Immunology and Allergology, Biology, “Victor Babes” University of Medicine and Pharmacy, 300041 Timisoara, Romania; geomed88@gmail.com (C.A.T.); vpaunescu@umft.ro (V.P.); 2Center for Gene and Cellular Therapies in the Treatment of Cancer—Oncogen Center, Clinical County Hospital “Pius Brînzeu”, 300723 Timisoara, Romania; mirabela.cristea@oncogen.ro (M.I.C.); teleaada@gmail.com (A.T.); camelia.oprean@umft.ro (C.O.); 3Department of Drug analysis, Chemistry of the Environment and Food, “Victor Babes” University of Medicine and Pharmacy, 300041 Timisoara, Romania; 4Translational and Experimental Clinical Research Centre in Oral Health, Department of Preventive, Community Dentistry and Oral Health, “Victor Babes” University of Medicine and Pharmacy, 300041 Timisoara, Romania

**Keywords:** stem cells, human exfoliated deciduous teeth, SHED, lipid metabolism

## Abstract

The metabolic regulation of stemness is widely recognized as a crucial factor in determining the fate of stem cells. When transferred to a stimulating and nutrient-rich environment, mesenchymal stem cells (MSCs) undergo rapid proliferation, accompanied by a change in protein expression and a significant reconfiguration of central energy metabolism. This metabolic shift, from quiescence to metabolically active cells, can lead to an increase in the proportion of senescent cells and limit their regenerative potential. In this study, MSCs from human exfoliated deciduous teeth (SHEDs) were isolated and expanded in vitro for up to 10 passages. Immunophenotypic analysis, growth kinetics, in vitro plasticity, fatty acid content, and autophagic capacity were assessed throughout cultivation to evaluate the functional characteristics of SHEDs. Our findings revealed that SHEDs exhibit distinctive patterns of cell surface marker expression, possess high self-renewal capacity, and have a unique potential for neurogenic differentiation. Aged SHEDs exhibited lower proliferation rates, reduced potential for chondrogenic and osteogenic differentiation, an increasing capacity for adipogenic differentiation, and decreased autophagic potential. Prolonged cultivation of SHEDs resulted in changes in fatty acid composition, signaling a transition from anti-inflammatory to proinflammatory pathways. This underscores the intricate connection between metabolic regulation, stemness, and aging, crucial for optimizing therapeutic applications.

## 1. Introduction

Mesenchymal stem cells (MSCs) are defined as self-renewing, progenitor cells that can differentiate into multiple specialized cell types including bone, cartilage, muscle, and fat cells [1]. Initially isolated from the bone marrow, MSCs were originally identified as supportive cells facilitating hematopoiesis within the intricate microenvironment of the bone marrow. As research has progressed, their versatility has become increasingly apparent, extending beyond their hematopoietic support role to encompass a broader spectrum of regenerative and therapeutic applications [2]. However, it has been subsequently discovered that numerous other tissues and organs, such as adipose tissue, umbilical cord, placenta, and teeth, are rich sources of MSCs, and thus represent alternative sources for their isolation [3,4]. Dental pulp stem cells (DPSCs) in permanent teeth and stem cells from human exfoliated deciduous teeth (SHEDs) are known to have MSC-like features, including a fibroblast-like morphology and adherence to plastic surfaces in cell culture experiments, and share a variety of cell surface markers, including CD90, CD29, CD44, CD73. DPSCs and SHEDs lack hematopoietic markers expression: CD14, CD19, CD34, and CD45 [4,5,6,7,8,9,10,11]. DPSCs and SHEDs can be easily isolated and expanded in vitro from teeth that would otherwise be discarded as medical waste. The collection of SHEDs does not involve invasive procedures to donors, as the teeth are already naturally exfoliated. This characteristic makes SHEDs an attractive and ethically acceptable source of stem cells for research and potential therapeutic applications, contributing to their growing importance in regenerative medicine and various biomedical studies [12]. Nevertheless, their potential as stem cell therapies in regenerative medicine studies has been less extensively investigated when compared to MSCs derived from bone marrow or adipose tissue, and their utilization in tissue engineering research has garnered interest due to their lower ethical and legal restrictions [13]. More recent publications suggest the potential utility of SHEDs towards a spectrum of conditions, including osteoarthritis [14,15], wound healing [16], hair regeneration [17], cardiovascular diseases (CVDs) [18], nervous system injury [19], Parkinson’s disease [20], and type 2 diabetes [21].

Compared to DPSCs, SHEDs are known to have a higher proliferation rate, increased cell population doubling, sphere-like cell-cluster formation capability, and multidifferentiation capacity [22,23,24,25,26]. These features make SHEDs a promising candidate for tissue replacement therapies. At the same time, the studies performed by Bhandi et al. revealed that SHEDs exhibit a greater osteogenic potential compared to DPSCs. This heightened potential is linked to elevated expressions of PTH (parathyroid hormone 1) and PTH1R (parathyroid hormone 1 receptor). These factors play a regulatory role in key osteogenic factors such as BMP, TGF-β, and IGF-14. A decline in osteogenic potential and a corresponding increase in adipogenic potential were observed in DPSCs compared to SHEDs, suggesting an age-related attenuation of osteogenesis in favor of adipogenesis [27]. In humans, the transition from deciduous teeth to adult permanent teeth is a dynamic and irreversible process characterized by a meticulously orchestrated sequence of events. This intricate progression involves the sequential release of chemical mediators that intricately coordinate the resorption of the roots of deciduous teeth, marking a crucial phase in dental development. This finely tuned mechanism reflects the intricate interplay of molecular signals and cellular activities governing the complex and precisely timed transition from primary to permanent dentition [28]. 

The study aims to examine the functional traits of SHEDs. In order to scrutinize the gradual transitions accompanying the transformation from deciduous to permanent teeth, SHEDs were isolated and cultured from incisors, canines, and molars of naturally exfoliated human deciduous teeth sourced from children aged 7 to 12 years. Comprehensive immunophenotypic and functional assessments were conducted from early passages through passage 10. Our analysis revealed that all SHED populations expressed the MSC markers like CD90, CD29, CD44, CD73, and variable levels of CD105, CD56, and CD117, and none of the hematopoietic markers CD34 or CD45. The dynamic expression of markers and growth kinetics were monitored from early passages up to passage 10. Additionally, the maintenance of autophagic capabilities was evaluated. Cell plasticity and differentiation capacity towards osteogenic, chondrogenic, adipogenic, and neuronal lineage were evaluated. Fatty acids composition including omega and nonessential fatty acids, but also lipid profiles indicative of cellular potency and aging, were assessed at different passages. The fatty acid composition of cells can have significant impacts on their structure and function, affecting cellular processes such as signaling pathways, gene expression, and membrane fluidity [29]. Since lipids have crucial roles in various cellular functions, exerting profound influences on essential cellular processes accountable for stem cell maintenance, differentiation, and aging, the lipid profile might represent a potent and promising future approach for cell function characterization [30,31,32]. Exploring and understanding the intricate interplay of lipid components within cells not only enhances our insights into fundamental cellular mechanisms but also opens venues for innovative approaches in future cell function assessments.

## 2. Results

### 2.1. Growth Kinetics of SHEDs

A total of 24 deciduous teeth were collected for SHEDs isolation, comprising eleven incisors, seven canines, and six molars. The overall success rate for SHED recovery was 79%, with adherent cells observed in culture dishes for up to 21 days after seeding. Multiple clonogenic and proliferative populations of SHED were established, and cells were passaged after sufficient expansion. The SHED populations were maintained in culture for up to 10 passages, with the same number of cells being used to seed a new plate at each passage. The growth kinetics of SHEDs from P1 to P10 was assessed (Figure 1A). The growth curves of all SHEDs show that the canines grew the slowest. Based on our calculations of the cell population doubling time, the cell PDT of the SHEDs isolated from incisors was 45.22 ± 7.56 h, for those isolated from canines was 65.44 ± 16.94 h, and for those isolated from molars was 59.77 ± 12.87 h (Figure 1B).

The significant differences observed in the growth kinetics of incisors compared to canines and molars might be explained by the chronological age at which the transition from deciduous to permanent dentition takes place. The eruption of permanent teeth is a gradual process that occurs over several years and involves two stages. The first active stage occurs between the ages of 6 and 8 years old, and during this stage, the permanent first molars will erupt behind the primary molars, which are still in place, followed by the beginning of the exfoliation of the primary teeth as the permanent incisors are being replaced. The first permanent molars usually erupt around the age of 6, while the incisors begin to erupt at around 7 years of age. The first active stage is followed by a silent period of 1.5–2 years before the second active stage begins. The second active stage occurs between the ages of 10 and 13 years old. This stage marks the completion of the exfoliation of deciduous canines and molars as they are replaced by permanent canines and premolars (ages of 10 and 11), together with the eruption of the second permanent molars [33,34]. Nevertheless, tooth eruption is a complex process that can be influenced by a multitude of factors such as genetics, nutrition, hormonal factors, gender differences (dental maturity generally being more advanced in females than in males in permanent dentition, in reverse of those for the deciduous dentition), and craniofacial morphology, which were beyond the scope of this study.

### 2.2. Multiparametric Immunophenotyping Analysis of SHEDs

During each passage, flow cytometry was employed to evaluate the immunophenotype of the cells and analyze the expression of MSC cell surface markers. While no unique MSC markers have been identified so far, we observed that the proliferating cells expressed the MSC markers CD90, CD29, CD44, CD56, CD73, CD105, and CD117, but not the hematopoietic stem cell-related molecules CD34 and CD45 [4,5,6,7,8,9,10,11]. Throughout the experimental period (P1–P10), the analyzed cell populations exhibited consistently high expression (over 70%) of CD90, CD29, CD44, and CD73 markers, as depicted in Figure 2A–D (and Appendix A). The expression pattern of CD105, CD56, and CD117 was found to be different. CD105 had a low expression level at the beginning of the cultivation period, with an average of only 8% positive cells, but significantly increased towards passage 10, reaching up to 48% (Figure 2E). Studies have shown that CD105 expression may be involved in increasing the chondrogenic potential [35] while decreasing osteogenic potential in some types of MSCs [36]. However, CD105 expression is debatable since the expression of this marker has been diversely reported in the literature at high expression levels (98%) [37,38], lower levels of CD105 (60.84% or 33.57%) [39], or very low levels (0.1%) [40]. The observations prompt inquiries into whether variations in isolation and experimental conditions could account for the phenotypic differences observed, or if other factors must be considered to account for the variability. CD56, a cell adhesion molecule, also known as Neural Cell Adhesion Molecule (NCAM), is not a typical marker found on MSC populations. However, it is widely expressed in tissues of neurogenic origins, controlling intercellular interactions and neurite growth, as well as neuronal differentiation, proliferation, and migration [41]. The expression of CD56 revealed an intriguing pattern in the SHED cell populations that we analyzed. We observed that the isolated SHED populations had CD56 molecule expression at around 32% at passage 1, which was slowly decreasing and eventually was lost by passage 9 (Figure 2F). It is not clear whether the loss of CD56 is due to the in vitro cell culture conditions or a normal biological phenomenon that occurs in vivo and is related to the aging of the SHEDs. CD56 expression on SHEDs might be related to the neural crest-cell origin of the dental pulp and, at the same time, SHEDs. Previous studies have reported the presence of CD56 in various dental cells and MSCs. In the mouse molar dental pulp, CD56 was identified in the central area by Obara and Takeda in 1993, but the specific cell types expressing this protein were not reported [42]. In 2008, Degistirici et al. isolated a group of progenitor cells derived from the neural crest of the third molar and found that they expressed CD56 [43]. Battula and colleagues reported in 2009 that a unique subset of MSCs from bone marrow expressed CD56 on their cell surfaces, but a different monoclonal antibody with a different epitope specificity was used to identify a CD56 isoform [44]. In 2010, Alipour et al. reported CD56 expression on SHEDs [45]. However, the functional significance of CD56 expression in SHEDs is still unknown. CD117 (c-kit) is a well-defined marker of stem cells, but it is also expressed on certain MSC subsets and has been reported to be expressed in adult/neonatal dental pulp cells [46,47], and periodontal ligament cells [48,49]. CD117 serves as a crucial growth factor receptor for Stem Cell Factor (SCF), which plays an important role in cell proliferation, cell migration, angiogenesis, and tissue remodeling [50,51]. Due to its involvement in the regulation of stem cell activity, CD117 is widely recognized as a key marker for identifying stem/progenitor cells. Our analysis showed that all SHEDs expressed CD117, confirming its presence in dental stem cells (Figure 2G). The expression of CD117 was initially lower, with 22% of cells on average being positive for this marker at the beginning of cultivation. However, its expression significantly increased during the cultivation period, reaching around 53% by passage 4. The role of CD117 in SHEDs might be associated with an enhanced capacity for proliferation and differentiation. Overall, the percentage of CD105, CD56, and CD117 positive cells was lower than other markers we examined in this study. 

### 2.3. In Vitro Plasticity Assessment of SHEDs

To test the in vitro plasticity of SHEDs, the cells were cultured in osteogenic, adipogenic, and chondrogenic induction medium for specific time intervals. The ability of SHEDs to undergo osteogenic, adipogenic, and chondrogenic differentiation was observed by confocal microscopy analyzing osteocalcin, aggrecan, and FABP-4 immunofluorescence staining of differentiated cells. The number of osteocalcin- and aggrecan-positive cells was high at P3, and maintained an increasing expression at P7, but gradually decreased by P10, suggesting a decreasing ability to differentiate into osteogenic and chondrogenic pathways (Figure 3A). The percentage of FABP-4 positive cells was reduced at P3, indicating a diminished adipogenic differentiation capacity of SHEDs, with our observation being in accordance with Miura et al. 2003, who observed that only 5% of cultured SHEDs were found to possess the potential to develop into Oil red O-positive lipid-laden fat cells [25]. However, the expression of the adipogenic marker, FABP4 increased towards P7 indicating a shift in the favor of adipogenic differentiation observed previously in aging stem cells [27,52] (Figure 3B). 

It is known that SHEDs express neuronal and glial cell markers like nestin, beta-III tubulin, GFAP, and other neural markers related to the neural crest-cell origin of the dental pulp [25,53]. We observed that cultured SHEDs expressed a variety of neural cell markers including nestin, βIII-tubulin, and GFAP, as detected by immunofluorescence staining (Figure 4). The expression of these markers was maintained during cultivation of SHEDs even at higher passages. However, a loss of integrity of cytoskeletal intermediate filaments of nestin (neuroepithelial stem cell protein) was observed at passage 10. Nestin is expressed in dividing cells during the early stages of development in the central nervous system (CNS) and peripheral nervous system (PNS), as well as in myogenic and other tissues. During cellular differentiation in vivo, the expression of nestin is transitory and gradually decreases, and tissue-specific intermediate filament proteins take over its function [52].

After 21 days of neural inductive cell culture, expression levels of neuronal markers were not significantly increased, but SHEDs lost part of their fibroblastic morphology and developed elongated cell-cytoplasmic processes (Figure 5). These traits remained consistent as the cells progressed through subsequent passages. Although a gradual decrease in GFAP and nestin expression was noted at P10, the cells maintained their elongated cell-cytoplasmic processes that were observed in earlier passages. 

### 2.4. SHEDs Fatty Acids Content Assessment

In this study, we aimed to investigate the changes in the fatty acid profile of SHEDs during in vitro cultivation. Several factors were taken into consideration, such as donor age, tooth type used for cell isolation, cell passage, and fatty acid composition of the culture medium. Using gas chromatography, we analyzed the fatty acid profiles of SHEDs at different passages. Our results showed that the early passages of SHEDs contained a high percentage of omega-3 fatty acids (70%) (Figure 6). This percentage decreased only slightly during cultivation, reaching 62% at passage 9. However, this difference was not statistically significant. On the other hand, the percentage of omega-9 fatty acids in SHEDs increased from 12% in early passages to 15.5% at passage 9 (Figure 6B). Additionally, SHEDs acquired a small percentage (2.8%) of omega-6 fatty acids during cultivation. Nonessential fatty acids (non-EFAs) accounted for 19% of the total fatty acid composition during cultivation. 

The GC–MS analysis of SHEDs revealed a high content of saturated fatty acids, which remained relatively stable throughout cultivation. In contrast, the unsaturated fatty acid content increased slightly from 12% in early passages to 18% in the ninth passage, but these differences were not statistically significant (Figure 7A) (see also Appendix A). Interestingly, by passage 9, two additional fatty acids were identified in the unsaturated fatty acid profile: oleic acid (18:1 cis-9) and arachidonic acid (20:4n-6) which accounted for 12% and 2%, respectively. These changes in fatty acid composition can have implications for the production of proinflammatory eicosanoids in aging cells [54]. At the same time, the decrease in elaidic acid (18:1) and saturated palmitic acids (16:0) could be indicative of changes associated with cellular senescence and aging (Figure 7B). Notably, the analysis revealed no significant differences in fatty acid composition between the tooth types used for SHEDs isolation (incisors, canines, and molars). 

Overall, the GC–MS analysis of SHEDs’ fatty acid composition provides insight into potential mechanisms underlying the cellular processes associated with cellular senescence and aging. Studies have shown that the balance of fatty acids in stem cells can affect their ability to differentiate and maintain their pluripotency. Stearate and palmitate are both major components of the fatty acid composition of stem cells, and stearoyl-CoA desaturase plays an important role in converting stearate into oleate, a preferred substrate for lipid synthesis. The incorporation of polyunsaturated fatty acids, especially omega-6, can increase membrane fluidity and flexibility, which are important for cellular function [55]. Arachidonic acid, which is found in cell membranes, can also contribute to membrane flexibility, but it can also be oxidized to produce proinflammatory resolving mediators [56,57]. Therefore, the balance of fatty acids in stem cells is an important parameter for their maintenance and function.

### 2.5. Autophagy Detection

The LC3B protein plays a central role in autophagy. Normally, this protein resides in the cytosol, but following cleavage and conjugation with phosphatidylethanolamine, LC3 associates with the phagophore and serves as a marker of autophagosomes. Among the four LC3 isoforms, LC3B is the most commonly used. Initially, LC3 is processed by Atg4, resulting in LC3-I. LC3-I is then conjugated with phosphatidylethanolamine (PE) to form LC3-II. LC3-II is associated with both the outer and inner membranes of the autophagosome. Following fusion with the lysosome, LC3 is cleaved from the outer membrane, while lysosomal enzymes degrade LC3 on the inner membrane, leading to a low LC3 content in the autolysosome. Consequently, endogenous LC3 can be visualized via fluorescence microscopy as a diffuse cytoplasmic pool or structures, primarily representing autophagosomes [58]. Across all assessed passages, SHEDs exhibited a basal level of autophagy, and LC3-positive autophagosomes were consistently present in different quantities (Figure 8A). However, there was a significant decrease in LC3B intensity fluorescence associated with autophagosome numbers towards passage 9, compared to passage 6. A significant loss of vimentin structure was observed in certain cells, particularly towards passage 6 and even more prominently towards passage 9. Upon exposure to B chloroquine diphosphate for 16 h, a distinctive significant increase in the number of autophagosomes associated with heightened LC3B expression was observed in all passages (Figure 2B). This suggests that normal autophagic flux was disrupted, leading to autophagosome accumulation. Normally, cellular autophagic activity is low under basal conditions but is markedly upregulated by exposure to B chloroquine diphosphate.

## 3. Discussion

Human SHEDs, first isolated by Miura et al. in 2003 from the dental pulp of exfoliated deciduous teeth, manifest a high proliferative potential, self-renewal capacity, and multilineage differentiation [25]. Based on their properties and origin from young donors (6–12 years old), SHEDs are considered a more juvenile form of stem cells than those obtained from permanent teeth. The results of the present study showed that SHEDs have a marked capacity to proliferate, steadily expressing high levels of CD73, CD90, CD29, and CD44. However, other markers’ expression was altered during the in vitro culture and propagation, with CD105, and CD56 molecules being the most representative in this regard. 

During long-term culture of SHEDs, the number of CD105-positive cells was found to be significantly increased, which is consistent with the findings of a study on amniotic fluid MSCs conducted by Wang et al. in 2020 [59]. These results suggest that in vitro cultivation may lead to the positive selection of CD105-expressing cells or induce the expression of CD105 on previously negative cells. In a previous study, Wang et al. (2014) also observed that the expression of CD105 was associated with decreased proliferation potential due to cell cycle arrest in the S-phase, a finding that is similar to our observation that CD105 expression was highest in late-passage cells, which had a slower proliferation rate compared to early-passage cells [60]. Changes in expression patterns were also observed for CD56. In contrast to some previous studies by Ducret et al. 2016, we observed a consistent and gradual decrease in CD56 expression [61]. This could be due to the preferential expansion of a specific CD56 negative dental stem cell population, or a de facto decrease of the CD56 expression in the CD56-positive population. While the significance of changes in CD105 and CD56 expression remains a subject of debate, our findings might suggest that MSCs, including SHEDs, are heterogeneous and may contain multiple subpopulations with varying proliferative and differentiation potentials. Nevertheless, SHEDs possess significant potential for differentiating into multiple lineages, including osteogenic, chondrogenic, and adipogenic lineages, as well as express several neural markers indicating their neural origin and neurogenic differentiation capability. SHEDs express their multilineage differentiation capacity during in vitro cultivation. However, it is worth noting that this capacity may diminish over time as these cells age. Extended periods of cell culture and repeated passaging have been demonstrated to induce cellular senescence, a phenomenon observed in various cell types, including MSCs (7).

Lipids play a critical role in stem cell differentiation. Research indicates that adult stem cells exhibit a metabolic profile similar to other quiescent cells, characterized by decreased glutamine consumption, reduced biosynthesis of nucleotides and lipids, and increased levels of fatty acid oxidation and ROS detoxification [62]. While glucose is the primary energy source for proliferating stem cells, quiescent and differentiating stem cells require additional energy substrates such as fatty acids and amino acids to maintain homeostasis and promote differentiation [63]. Fatty acid synthesis is initiated with the formation of palmitic acid (C16) from acetyl-CoA and malonyl-CoA. Unlike fatty acid catabolism, which predominantly occurs in the mitochondria, fatty acid synthesis takes place in the cytoplasm. The fatty acid synthase system, which is made up of several enzymes connected by an acyl carrier protein (ACP), is present in the cytoplasm, along with its substrates [64]. Cytoplasmic acetyl-CoA is predominantly derived from mitochondrial acetyl-CoA, thanks to the citrate–malate shuttle. This shuttle facilitates the transfer of acetyl groups from the mitochondria, where deacetylation occurs, to the cytosol, where acetylation takes place. In the cytosol, acetyl-CoA and malonyl-CoA join with the synthase and ACP. Subsequently, a series of seven acetyl group transfers occur, culminating in the formation of palmitoyl-ACP. Finally, palmitic acid is liberated from palmitoyl-ACP. Palmitic acid serves as a precursor for several long-chain fatty acids, including stearic acid, palmitoleic acid, and oleic acid. Generally, elongation or desaturation steps follow, for example, desaturation at C9, to produce oleic acid from stearic acid [64,65]. Membrane glycerophospholipids, such as phosphatidylcholine (PCs) and phosphatidylethanolamine (PEs), are hallmarks of senescence and were found to increase in aged MSCs [66]. Another significant metabolic process that changes with aging is the conversion of saturated fatty acids (SFA) to mono-unsaturated fatty acids (MUFA) [66]. Our study reveals that the unsaturated fatty acid content in older SHEDs increased slightly from 12% in early passages to 18% in the ninth passage. Interestingly, we observed a similar shift in fatty acid composition to that seen in aging bone marrow MSCs, which confirms one of the central features of aging MSCs observed by Kilpinene et al. 2013, the gain of arachidonic acid (20:4n-6a) [54]. Arachidonic acid is a precursor for several pro- and also anti-inflammatory molecules, and it is plausible that the production of 20:4n-6-a-derived proinflammatory eicosanoids is facilitated in aging cells through this pathway [67,68,69]. Along with arachidonic acid (20:4n-6), the increase of individually saturated fatty acid 18:0 (stearic acid) concentration was also observed in the late passages, indicating a shift in the lipid signaling pathway from an anti- to proinflammatory direction, specific for senescent cells [54]. Our study indicated another mono-unsaturated omega-9 fatty acid, oleic acid (18:1 cis-9), accumulated (12%) to the unsaturated fatty acid pool by passage 9, while the concentration of the saturated fatty acid palmitic acid (16:0) slightly decreased. This phenomenon is also associated with senescence and can be linked to a marked decrease in the fatty acid synthase and stearoyl-CoA-desaturase 1 (also known as delta 9 desaturase) enzymes, which convert saturated fatty acids (such as palmitic and stearic acid) into their monounsaturated forms, including palmitoleic acid (16:1) and oleic acid (18:1), as reported by Nakamura and Nara in 2004 [70]. Despite the sustained levels of palmitic acid, the conversion of saturated fatty acids into monounsaturated fatty acids, particularly palmitic acid, is being ceased by decreasing or shutting down the synthesis of stearoyl-CoA-desaturase, blocking a metabolic pathway with an essential role in this conversion process [54,71,72]. Aging has been shown to impact various metabolic parameters, including lipid homeostasis. In humans and other organisms, aging is often associated with increased fat accumulation and alterations in membrane lipid structure that lead to reduced membrane fluidity [73,74]. With age, there is an increase in the level of unsaturation in membrane phospholipids, leading to the generation of more lipid peroxidation products, which can contribute to cellular damage [75]. 

SHEDs have the ability to perform autophagy and express a basal level of LC3B associated with autophagosomes during cultivation. Under normal circumstances, autophagy is only minimally activated, but it can be dynamically induced in response to cellular stressors such as nutrient deprivation and it helps maintaining cellular homeostasis [76]. Autophagy is an active process that plays a critical role in the metabolic state of cells and is essential for the survival of long-lasting stem cells, particularly in the context of aging and degenerative conditions that affect stem cell regenerative potential [77]. With aging, autophagy becomes increasingly important for maintaining regenerative potential [78]. LC3B antibody analysis revealed a low basal level of autophagy in SHEDs at all passages evaluated, with a slight decrease observed during later passages when some cells also exhibited a loss of vimentin. Reduced autophagy can lead to a breakdown in proteostasis, accumulation of misfolded and aggregated proteins, and an increase in oxidative stress in MSCs [77]. Maintaining a balanced proteome is a challenging task for the cell, especially in the presence of various external and endogenous stressors that accumulate with aging cells [79]. 

Further investigations are necessary to understand the impact of changes in fatty acid composition during SHEDs cultivation and to optimize culture conditions for preserving their advantageous features. A comprehensive understanding of stem cell metabolism during in vitro cultivation and differentiation is essential to maximize their regenerative potential while promoting proliferation without undermining their ability to differentiate. The existing methods employed for in vitro expansion of MSCs present limitations that hinder their widespread use, notably impacting multilineage differentiation potential during monolayer culture [7,80]. Addressing these challenges is crucial for optimizing stem cell metabolism and achieving enhanced functionality in therapeutic applications. Although progress has been made, a complete understanding of stem cell metabolism remains elusive. However, achieving this knowledge could lead to significant advancements and progress in the manipulation and application of stem cells in clinical medicine.

## 4. Materials and Methods

### 4.1. Isolation of SHEDs and Ethical Approval

Normal exfoliated human deciduous incisors, canines, and molars were collected from 12 children, aged 7 to 12 years old, after informed consent was obtained from the parents or legal tutors. The pulp was separated from the remnant crown, and SHED cells were obtained by outgrowth from tissue explants. The cells were grown in Minimum Essential Medium (MEM) alpha cell culture medium (Gibco, Life Technology, Paisley, UK), supplemented with 10% fetal calf serum (FCS; PromoCell, Heidelberg, Germany) and l% penicillin–streptomycin (Gibco, Life Technology Corporation, Grand Island, NY USA). The medium was renewed twice a week and subcultures were performed at 90% confluence. The cells were always plated at the same cellular density during their in vitro cultivation (3.5 × 10^5^ cells/75 cm^2^ flask).

All procedures involving human participants were in accordance with the ethical standards of the institutional research committee and with the 1964 Helsinki Declaration and its later amendments or comparable ethical standards. Informed consent was obtained from all individual participants included in the study. For each experiment, we analyzed at least three biological replicates.

### 4.2. Growth Kinetics

Growth kinetics of SHED from P1 to P10 was assessed. SHEDs were inoculated on a T75 culture flask at a density of 3.5 × 10^5^ cells/flask, and the cells were counted 7 days after plating when they reached 90% confluency. The growth kinetics was calculated using the following formula:$$\mathrm{Growth}\;\mathrm{kinetics}=T\times \frac{ln2}{ln\left(\frac{\mathrm{Fc}}{\mathrm{Ic}}\right)}$$
where T is the cell culture time, Ic is the initial number of cells, and Fc is the final number of cells [7,81].

### 4.3. Multiparametric Immunophenotyping of SHEDs

Cell suspensions were prepared and 1 × 10^6^ cells were incubated for 25 min at 4 °C in the dark with a panel of antibodies, as presented in Table 1. Flow cytometry was performed using a four-color FACSCalibur system (BD Biosciences, San Jose, CA, USA). Unstained controls were also used to adjust for the background and cell autofluorescence. The experiment was repeated at least three times. The single cell population was identified by defining the gated population on a side scatter area signal vs. a forward scatter area (SSC-A/FSC-A) signal dot plot. Analysis was performed using the Cell Quest Pro software, version 6.0 (BD Biosciences, San Jose, CA, USA).

### 4.4. In Vitro Plasticity of SHEDs

In vitro trilineage differentiation potential of SHEDs was tested using a human mesenchymal stem cell functional identification kit (R & D Systems, Minneapolis, MN, USA, SC006) according to the manufacturer’s instructions. The cells reaching passages 3, 7, and 10 were induced towards osteogenic, chondrogenic, and adipogenic lineages as indicated, using the media supplements included in the kit. The osteogenic supplement, a 20× concentrated solution within the kit, included dexamethasone, ascorbate–phosphate, and β-glycerolphosphate—sufficient for supplementing 50 mL of medium. The chondrogenic supplement contained 0.5 mL of a 100× concentrated solution with dexamethasone, ascorbate–phosphate, proline, pyruvate, and recombinant TGF-β3, enough for 50 mL of medium. For adipogenic differentiation, the kit included 0.5 mL of a 100× concentrated solution containing hydrocortisone, isobutylmethylxanthine, and indomethacin in 95% ethanol—adequate for supplementing 50 mL of medium. SHED cells were plated on IBIDI (IBIDI, Gmbh, Martinsried, Germany) tissue-culture dishes for 14 days with osteogenic, 21 days with adipogenic, and 28 days with chondrogenic differentiation medium, respectively. Controls and differentiated cells were fixed with 4% paraformaldehyde for 20 min at room temperature and stained using goat anti-mouse FABP-4 antigen affinity-purified polyclonal antibody (adipocytes), a goat anti-human aggrecan antigen affinity-purified polyclonal antibody (chondrocytes), and a mouse anti-human Osteocalcin monoclonal antibody (osteocytes) for the confirmation of differentiation status. Neural differentiation induction was based on a modified protocol that has been previously reported for neural differentiation of multipotent adult progenitor cells [82]. The cells were cultivated in DMEM/F12 (Sigma-Aldrich, Ayrshirf, UK) supplemented with 1% ITS (R & D Systems, Minneapolis, USA) and 100 ng/mL FGF (fibroblast growth factor) (R & D Systems, Minneapolis, USA) for 7 days, followed by 100 ng/mL bFGF (R & D Systems, Minneapolis, USA), 10 ng/m1 FGF8 (Sigma, F6926), and 100 ng/mL sonic hedgehog (SHH, Sigma, S0191) and ascorbic acid 200 µM for an additional 14 days. The cells were fixed with 4% paraformaldehyde for 20 min at room temperature and stained using goat anti-rat nestin polyclonal antibody (R & D Systems, Minneapolis, USA), mouse anti-neuron-specific beta-III tubulin monoclonal antibody (R & D Systems, Minneapolis, USA), and sheep anti-human Glial Fibrillary Acidic Protein (GFAP) polyclonal antibody (R & D Systems, Minneapolis, USA). The cells were analyzed using a Zeiss LSM 710 confocal microscope (Zeiss GmbH, Oberkochen, Germany).

### 4.5. SHEDs Fatty Acids Content Assessment

For each experimental variant, 5 × 10^5^ cells were repeatedly washed with ice-cold PBS, centrifuged at 1500 rpm for 10 min, and the resulting pellet was further used for lipid extraction. For lipid extraction, the cells were mixed with 400 mL chloroform/methanol 2:1 and centrifuged at 4000 rpm for 5 min. The chloroform phase was transferred into a clean tube and 200 mL of water and 100 mL of chloroform were added to the previous chloroform phase, mixed, and centrifuged again at 4000 rpm for 10 min. The chloroform phase was transferred, evaporated to dryness under a stream of nitrogen, and resuspended in 100 mL of chloroform/methanol 1:2. A total of 50 mL of the mixture was treated by saponification for 1 h at 65 °C, then neutralized with HCl and extracted with hexane, according to a previously described method. The organic phase was dried and resuspended in 200 mL methanol, followed by derivatization to fatty acid methyl esters and subsequent analysis by gas chromatography coupled with mass spectrometry (GC/MS) [83]. All solvents and reagents used for lipid extraction, derivatization, and analysis were GC-grade purity (Sigma-Aldrich, St. Louis, MO, USA). Sample analysis was performed on an Agilent 6890 gas chromatograph coupled with a 5973 MSD quadrupole mass spectrometer (Agilent Technologies, Santa Clara, CA, USA). Compounds separation was performed on an HP-5MS (Agilent) fused-silica capillary column (30 m × 0.25 mm × 0.25 μm) at constant high purity (6.0) helium flow of 1 mL/min. The column oven temperature was set at a rate of 6 °C/min, starting from a temperature of 50 °C with 1 min hold up to 300 °C with 5 min final hold. Samples were injected in a nonsplit mode, and the inlet temperature was set at 230 °C. The MS scan parameters included a full scan with a mass range of m/z 50–500 Da, and the ionization energy was set at 70 eV. Compounds were identified using NIST11 mass spectral library in Chemstation software, ver. B.01.00 (Agilent Technologies, Palo Alto, CA, USA). The mass spectra of the lipid molecules found in the samples were accurately identified by comparison to the mass spectra from the MS database library. Solvent blank and control samples were run under similar GC/MS conditions in order to exclude potential reagent or laboratory contamination.

### 4.6. Autophagy Detection

In order to evaluate autophagy, we used the LC3B antibody kit for autophagy, from Invitrogen (Molecular Probes, Live Technologies, Eugene, OR, USA). The kit included a rabbit polyclonal antibody for LC3B, and B chloroquine diphosphate to artificially induce autophagosome formation as a positive control for autophagy. The basal level of autophagy was assessed at different passages (P3, P6, and P9) during the cultivation of SHEDs. The staining of LC3B was carried out in accordance with the manufacturer’s instructions. Firstly, the cells were fixed with a 3.7% formaldehyde solution for 15 min at room temperature. After fixation, the cells were washed and permeabilized with 0.2 Triton X-100 in PBS. The diluted primary antibody (0.5 µg/mL working solution in 1% BSA in PBS) was added to the cells and incubated for 1 h. A goat anti-rabbit DyLight 550 secondary antibody (Invitrogen, Rockford, IL, USA) was used. In parallel, a primary mouse anti-vimentin antibody (Dako, Glostrup, Denmark) was used to stain the vimentin structure. Nuclei counterstaining was performed with DAPI (4′,6-diamidino-2-phenylindole) (Sigma-Aldrich, St Louis, USA). B chloroquine diphosphate was used to induce autophagy and to evaluate the dynamic process of autophagosome formation in order to distinguish between baseline levels of autophagy. The cells were washed twice with prewarmed PBS and subsequently treated with B chloroquine diphosphate, in accordance with the manufacturer’s instructions. The staining process was carried out using the same protocol as described earlier. Quantification of fluorescence was performed using Zen Imaging software, version 8,0,5,273 from Zeiss.

### 4.7. Statistical Analyses

One-way analysis of variance (ANOVA) was performed to assess differences between the experimental groups (incisors, canines, and molars), followed by pairwise post hoc t-tests using Microsoft Excel, Office Pro Plus, 2021. Parametric data are expressed as the means + standard deviation (SD). A value of *p* < 0.05 was considered statistically significant (* *p* < 0.05, ** *p* < 0.001, *** *p* < 0.0001).

## 5. Conclusions

SHEDs possess a remarkable capacity for multilineage differentiation, including osteogenic, adipogenic, and chondrogenic lineages, and exhibit neural markers that indicate their neural origin and neurogenic differentiation potential. With specific cultivation conditions, SHEDs can differentiate into bone, cartilage, neural, and adipose cells to varying extents. These attributes position them as promising candidates in the field of regenerative medicine and tissue engineering, where they can be employed either as a standalone cellular population or in conjunction with various biocompatible scaffolds. Nevertheless, like all cells, SHEDs will eventually reach their proliferative and regenerative limits in culture as they age. An understanding of their metabolism can facilitate the refinement of cultivation conditions, thereby preserving their valuable traits for more extended periods. 

To advance the field, forthcoming research endeavors might delve into the intricacies of MSCs metabolism. Investigating specific factors or signaling pathways that govern MSCs differentiation could unlock novel insights into enhancing their regenerative potential. Understanding the metabolic nuances of MSCs holds the key to refining cultivation conditions, thereby preserving their valuable traits for more extended periods and ensuring their continued efficacy in regenerative medicine and tissue engineering applications.

## Figures and Tables

**Figure 1 ijms-24-17249-f001:**
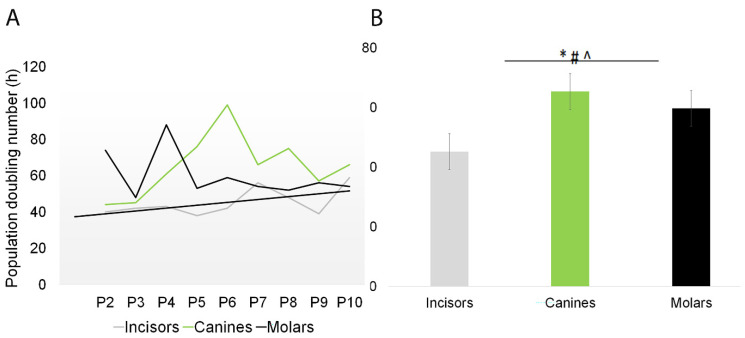
Growth kinetics of SHEDs. The number of SHEDs was counted each time following subculture from passages 2 to 10. The growth kinetics was calculated based on cell counts. (**A**) Population doubling time comparison of SHEDs from passages 2 to 10. (**B**) Comparative analysis of population doubling time in various SHED sources. The figure shows significant differences (*p* < 0.05) in population doubling time between incisors and canines (*), incisors and molars (#), and canines and molars (ˆ). The data summarize the findings from five individual experiments.

**Figure 2 ijms-24-17249-f002:**
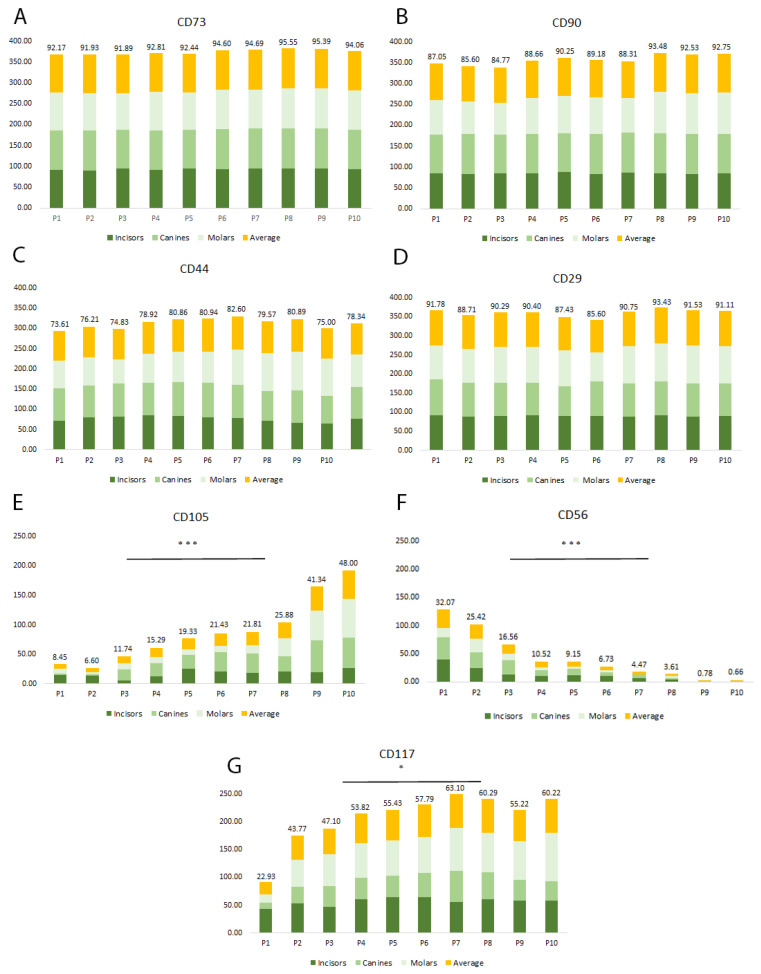
Flow cytometric analysis of MSC marker expression levels in proliferating SHED cells. Representative charts of flow cytometric analysis of average expression levels of MSC markers (CD90, CD29, CD44, CD73, CD105, CD56, and CD117 (**A**–**G**) in the proliferating SHED cells population. The figure highlights significant differences in several marker expression levels from passage 1 to passage 10 (* *p* < 0.05, *** *p* < 0.0001). The data summarize the findings from five individual experiments.

**Figure 3 ijms-24-17249-f003:**
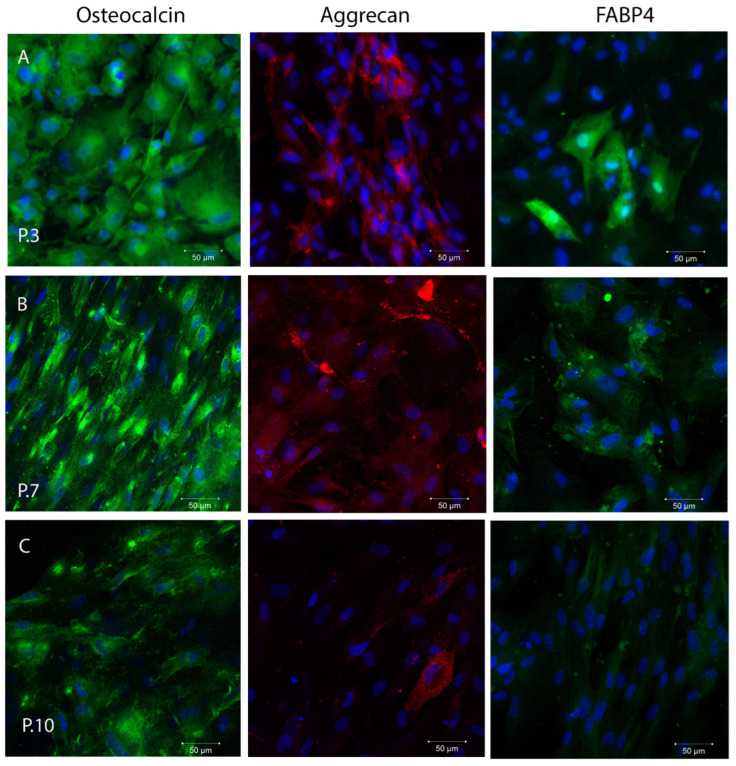
In vitro plasticity assessment of SHEDs revealed by immunofluorescence staining and confocal laser scanning microscopy, using a Zeiss LSM 710 confocal microscope, scale bar 50 µm, magnification 200×. In vitro osteogenic, chondrogenic, adipogenic differentiation of SHEDs at passage 3 (**A**), passage 7 (**B**), and passage 10 (**C**). Osteocalcin (green), aggrecan (red), FABP 4 (green). Nuclei counterstaining (in blue) was performed with DAPI. The experiments were repeated three times.

**Figure 4 ijms-24-17249-f004:**
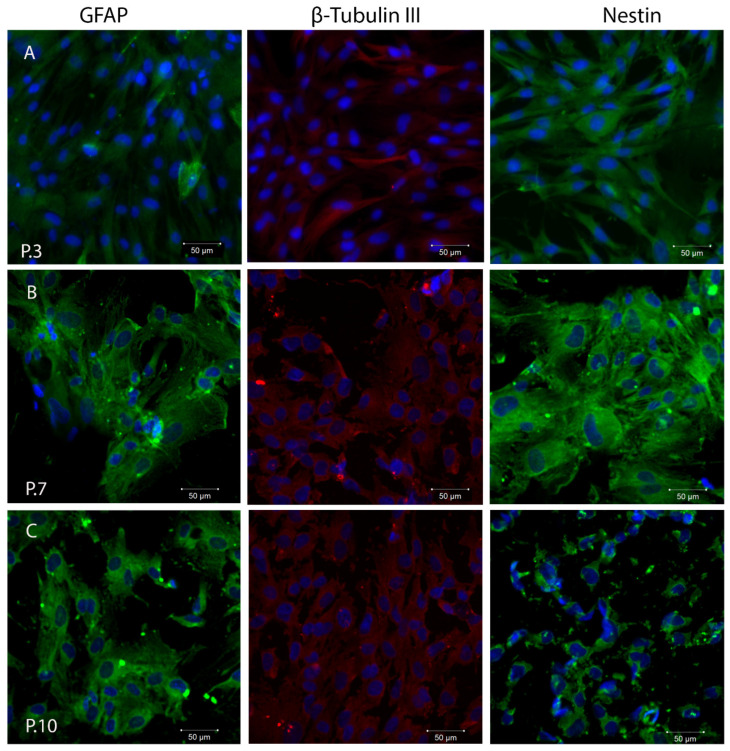
Evaluation of neural cell markers during SHEDs cultivation at passage 3 (**A**), passage 7 (**B**), and passage 10 (**C**), as detected by immunofluorescence staining and confocal laser scanning microscopy (Zeiss LSM 710 confocal microscope). Scale bar 50 µm, magnification 200×. GFAP (green), β-Tubulin III (red), and nestin (green), as detected by immunofluorescence staining. Nuclei counterstaining (in blue) was performed with DAPI. The experiments were repeated three times.

**Figure 5 ijms-24-17249-f005:**
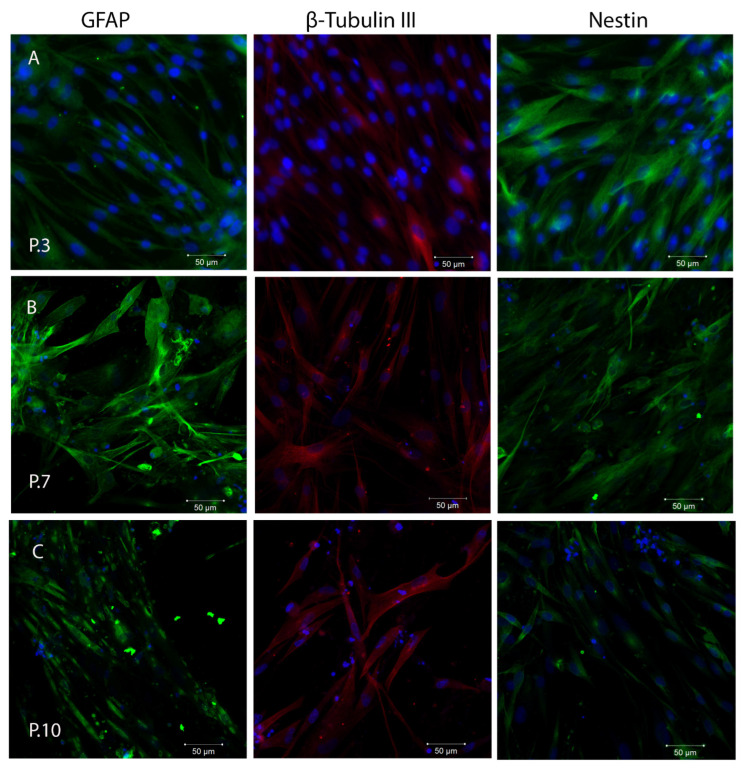
Neurogenic differentiation of SHEDs during cultivation at passage 3 (**A**), passage 7 (**B**), and passage 10 (**C**), as detected by immunofluorescence staining and confocal laser scanning microscopy, Zeiss LSM 710 confocal microscope. Scale bar 50 µm, magnification 200×. GFAP (green), β-Tubulin III (red), and nestin (green). Nuclei counterstaining (in blue) was performed with DAPI. The experiments were repeated three times.

**Figure 6 ijms-24-17249-f006:**
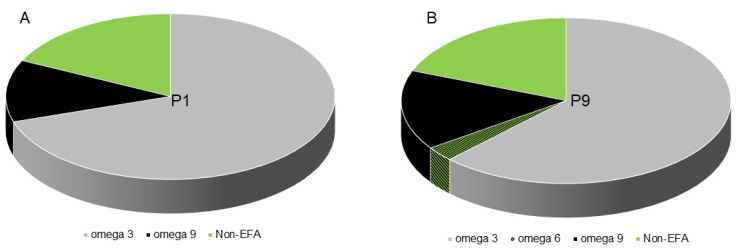
Representative charts of omega and nonessential fatty acids composition in SHEDs during in vitro cultivation. (**A**) Omega and nonessential fatty acids composition in SHEDs at passage 1. (**B**) Omega and nonessential fatty acids composition in SHEDs at passage 9. Data represent the results of three individual experiments.

**Figure 7 ijms-24-17249-f007:**
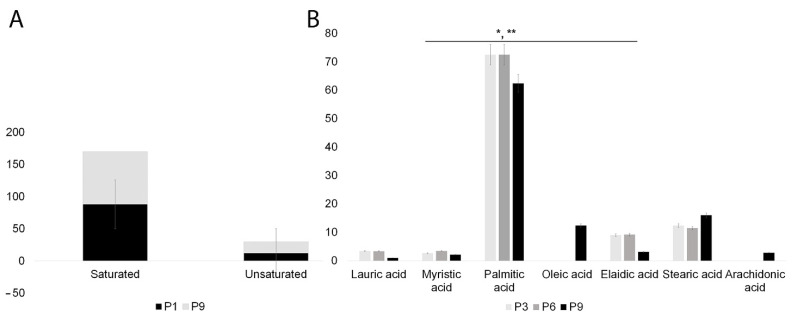
Representative charts of fatty acids composition and dynamics in SHEDs by GC–MS. (**A**) Saturated and unsaturated fatty acids composition in SHEDs passage 1 versus passage 9. (**B**) Variation of fatty acids composition during cultivation; by passage 9, unsaturated fatty acids profile was enriched with 12% oleic acid (18:1 cis-9) (**), whereas individual monounsaturated fatty acids like elaidic acid (18:1) and saturated palmitic acids (16:0) decreased significantly (*). Data represent the results of three individual experiments.

**Figure 8 ijms-24-17249-f008:**
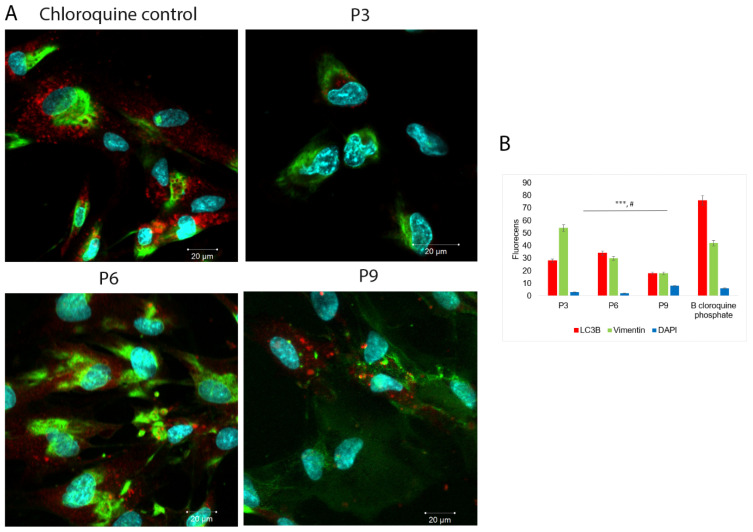
Detection of autophagic flux by confocal laser scanning microscopy in SHEDs at different passages during cultivation: passage 3, passage 6, and passage 9; scale bar 20 µm; magnification 630× (**A**). LC3 is recruited to autophagosomes forming round structures as indicated by red dots. The number of LC3-positive structures significantly increases following exposure to chloroquine diphosphate (**B**) (***). The vimentin structure is marked in green. A significant loss of vimentin structure was observed as the cells progress from passage 3 to passage 9 (#). Nuclei counterstaining (in blue) was performed with DAPI. The experiments were repeated three times.

**Table 1 ijms-24-17249-t001:** Fluorochrome-conjugated monoclonal antibodies used for immunophenotypic analysis.

Target	Fluorochrome	Clone	Manufacture	Isotype	Catalog Number
CD29	PE	MAR4	BD	Mouse BALB/c IgG1, k	555443
CD34	FITC	581	BD	Mouse IgG1, k	555821
CD44	FITC	L178	BD	Mouse BALB/c IgG1, k	347943
CD45	PE	HI30	BD	Mouse IgG1, k	555483
CD56	PE	B159	BD	Mouse IgG1, k	555516
CD73	PE	AD2	BD	Mouse IgG1, k	550257
CD90	FITC	5El0	BD	Mouse BALB/c IgG1, k	555595
CD105	FITC	266	BD	Mouse BALB/c IgG1, k	561443
CD117	PE	A3C6E2	Biolegend (San Diego, CA, USA)	Mouse IgG1k	323408

## Data Availability

Data are contained within the article or Appendix A.

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
