# Peer review of "Stem Cells Derived from Human Exfoliated Deciduous Teeth Functional Assessment: Exploring the Changes of Free Fatty Acids Composition during Cultivation"

_ijms, 2023, doi:10.3390/ijms242417249_

Round 1

Reviewer 1 Report

Comments and Suggestions for Authors

Although the study is well-designed, there is room for improvement in the quality of the manuscript. I have a few suggestions to enhance its overall quality.

1. Towards the end of the abstract, you mentioned that prolonged cultivation of stem cells from human exfoliated deciduous teeth (SHEDs) resulted in changes in fatty acid composition and a shift in the lipid signaling pathway. It would be beneficial to briefly highlight the potential implications or significance of these findings. For example, how might these changes impact the therapeutic applications of SHEDs or our understanding of their behavior in an aging context? Adding a sentence or two to address these points will provide a stronger conclusion to your abstract.

2. The introduction provides a good overview of mesenchymal stem/stromal cells (MSCs) and their various sources. However, it would be beneficial to briefly mention the potential applications of MSCs in regenerative medicine or tissue engineering to provide context for the importance of studying dental pulp stem cells (DPSCs) and SHEDs.

3. The sentence mentioning the studies performed by Bhandi et al. could benefit from a clearer explanation of the implications of their findings.

4. The "Introduction" section should be broken into several paragraphs. This could help provide a clearer structure and improve readability.

5. The equation used in the subsection "4.2. Growth kinetics" requires citation.

6. In the "Materials and methods" section, the number of replicates for each test should be specified.

7. The "Discussion" section also needs to be divided into several paragraphs. The current format is difficult to read and can make it challenging for readers to follow the logical progression of ideas. By breaking the discussion into paragraphs, each focusing on a specific aspect or finding, it will enhance the readability and organization of the section.

8. Towards the end of the discussion, you mentioned the importance of understanding stem cell metabolism during in vitro cultivation and differentiation to maximize their regenerative potential. Consider expanding on this point by highlighting specific challenges or areas of research that need further exploration to gain a comprehensive understanding of stem cell metabolism.

9. Most of the references are not up-to-date. The obtained results should be discussed using the most recently published papers.

10. The "Conclusions" section effectively summarizes the key points discussed in the previous sections. To further enhance the conclusion, you could consider adding a sentence or two about the potential future directions or areas of research that could build upon the findings discussed in the study. This could include investigating the role of specific factors or signaling pathways in SHED differentiation or examining their therapeutic potential in preclinical and clinical studies.

11. During my review, I noticed a few areas where the language and grammar could benefit from further improvement to enhance the clarity and impact of your research. To ensure the manuscript meets the highest standards of English language usage, I would like to suggest considering the assistance of a professional editor.

Comments on the Quality of English Language

During my review, I noticed a few areas where the language and grammar could benefit from further improvement to enhance the clarity and impact of your research. To ensure the manuscript meets the highest standards of English language usage, I would like to suggest considering the assistance of a professional editor.

Author Response

Ref. Manuscript ID: ijms-2729559

Answers to Reviewer 1

Reviewer 2 Report

Comments and Suggestions for Authors

Review of manuscript ID: ijms-2729559 entitled: Stem Cells Derived from Human Exfoliated Deciduous Teeth Functional Assessment: Exploring the Changes of Free Fatty Acids Composition During Cultivation’.

The above manuscript focuses on the regulation of stemness metabolism, which is now widely recognized as a crucial determinant of stem cell fate. When mesenchymal stem cells (MSCs) are transferred to a stimulating and nutrient-rich environment, they rapidly proliferate and undergo changes in protein expression, accompanied by a significant reconfiguration of central energy metabolism. The shift from quiescent to metabolically active cells can result in an increase in the proportion of senescent cells, which can limit their regenerative potential. In this study, human exfoliated deciduous teeth (SHEDs) were cultured in vitro for up to ten passages. Immunophenotypic analysis, growth kinetics, in vitro plasticity potential, fatty acid content, and maintenance of autophagic capacity were evaluated. Authors report that SHEDs express distinct cell surface markers, have a high self-renewal capacity, and possess a unique potential for neurogenic differentiation. Compared to younger SHEDs, older SHEDs exhibit lower proliferation rates, reduced potential for chondrogenic and osteogenic differentiation, increased capacity for adipogenic differentiation, and decreased autophagic potential. Additionally, the fatty acid composition of SHEDs shifted over time, indicating a change from an anti-inflammatory to a pro-inflammatory lipid-signaling pathway.

Overall, the manuscript requires revisions before being published.

Specific Points:

1.      The Introduction is short and lacking in terms of References. A minimum of 1,5 pages is required for the Introduction. There is plenty of literature on mesenchymal stem cells and their properties. See: PMID: 34115524 ; PMID: 37954462 ; PMID: 27527147 ; Most statements made in the Introduction are without References. Please revise and add appropriate references were needed.

2.      Figure 2 shows flow cytometric Analysis of MSC Marker Expression Levels in Proliferating SHED Cells. Authors need to show the actual flow cytometric data and not quantified graphs. Readers need to see these data. In addition, more validations are required to prove that the starting cells are MSCs. See Figure 1 and Figure 2 in https://doi.org/10.3390/ijms17081259

3.      A very important point: How is it that the MSCs are able to differentiate into neurogenic and adipogenic lineages over the passages (P1-P10) whilst maintaining the expression of MSCs markers such as CD73 and CD90?? This is impossible.

4.      The Introduction for section 2.4 must be included in the normal Introduction and not placed under results.

5.      The actual results and graphs from gas chromatography-MS must be displayed. Not just some quantification graphs (Figure 6).

6.      Figure 7 – show the actual results from gas chromatography-MS.

Comments on the Quality of English Language

Review of manuscript ID: ijms-2729559 entitled: Stem Cells Derived from Human Exfoliated Deciduous Teeth Functional Assessment: Exploring the Changes of Free Fatty Acids Composition During Cultivation’.

The above manuscript focuses on the regulation of stemness metabolism, which is now widely recognized as a crucial determinant of stem cell fate. When mesenchymal stem cells (MSCs) are transferred to a stimulating and nutrient-rich environment, they rapidly proliferate and undergo changes in protein expression, accompanied by a significant reconfiguration of central energy metabolism. The shift from quiescent to metabolically active cells can result in an increase in the proportion of senescent cells, which can limit their regenerative potential. In this study, human exfoliated deciduous teeth (SHEDs) were cultured in vitro for up to ten passages. Immunophenotypic analysis, growth kinetics, in vitro plasticity potential, fatty acid content, and maintenance of autophagic capacity were evaluated. Authors report that SHEDs express distinct cell surface markers, have a high self-renewal capacity, and possess a unique potential for neurogenic differentiation. Compared to younger SHEDs, older SHEDs exhibit lower proliferation rates, reduced potential for chondrogenic and osteogenic differentiation, increased capacity for adipogenic differentiation, and decreased autophagic potential. Additionally, the fatty acid composition of SHEDs shifted over time, indicating a change from an anti-inflammatory to a pro-inflammatory lipid-signaling pathway.

Overall, the manuscript requires revisions before being published.

Specific Points:

1.      The Introduction is short and lacking in terms of References. A minimum of 1,5 pages is required for the Introduction. There is plenty of literature on mesenchymal stem cells and their properties. See: PMID: 34115524 ; PMID: 37954462 ; PMID: 27527147 ; Most statements made in the Introduction are without References. Please revise and add appropriate references were needed.

2.      Figure 2 shows flow cytometric Analysis of MSC Marker Expression Levels in Proliferating SHED Cells. Authors need to show the actual flow cytometric data and not quantified graphs. Readers need to see these data. In addition, more validations are required to prove that the starting cells are MSCs. See Figure 1 and Figure 2 in https://doi.org/10.3390/ijms17081259

3.      A very important point: How is it that the MSCs are able to differentiate into neurogenic and adipogenic lineages over the passages (P1-P10) whilst maintaining the expression of MSCs markers such as CD73 and CD90?? This is impossible.

4.      The Introduction for section 2.4 must be included in the normal Introduction and not placed under results.

5.      The actual results and graphs from gas chromatography-MS must be displayed. Not just some quantification graphs (Figure 6).

6.      Figure 7 – show the actual results from gas chromatography-MS.

Author Response

Ref. Manuscript ID: ijms-2729559

Reviewer 3 Report

Comments and Suggestions for Authors

The paper's strengths include its evaluation of the dynamic expression and growth kinetics of markers, maintenance of autophagic capabilities, cell plasticity and differentiation capacity, and fatty acids composition and lipid profiles of SHEDs. Additionally, it discussed the isolation of SHEDs from normal exfoliated human deciduous incisors, canines, and molars collected from 12 children aged 7-to 12 years old and the in vitro plasticity assessment of SHEDs, which included the cells reaching passage 3, 7 and 10 being induced towards osteogenic, chondrogenic, and adipogenic lineages, and neural differentiation induction based on a modified protocol. Furthermore, the paper also discussed the statistical analyses of the data, the fatty acids content assessment of SHEDs, and the conclusions that SHEDs possess a remarkable capacity for multi-lineage differentiation, including osteogenic, adipogenic, and chondrogenic lineages, and exhibit neural markers that indicate their neural origin and neurogenic differentiation potential.

However, the paper did not discuss the ethical implications of using SHEDs in regenerative medicine and tissue engineering, nor did it discuss the potential safety risks of using SHEDs in clinical settings.

Comments on the Quality of English Language
  • The word “fibroblastlike” should be “fibroblast-like”.
  • The phrase “and share as well” should be “and share”.
  • "Marker's dynamic expression" should be "The dynamic expression of markers."

  • "The significant differences we observed" should be "The significant differences observed."

  • "Nerveless" should be "Nevertheless."

  • "During each passage, we employed flow cytometry" should be "During each passage, flow cytometry was employed."

  •  

Author Response

Ref. Manuscript ID: ijms-2729559

Answers to Reviewer 3

Reviewer 4 Report

Comments and Suggestions for Authors

The research question and the desired outcome should be clearly stated in the article. The main research question is not clearly and concisely presented in the article. However, upon closer examination of the article, it becomes clear that the research aims to investigate the functional properties of stem cells derived from human exfoliated deciduous teeth. This includes studying their proliferation, differentiation, and metabolic regulation. The presented submission also discusses the changes in the composition of free fatty acids during cultivation and their potential implications for stem cell-based therapies.

The present research focuses on stem cells derived from human exfoliated deciduous teeth, which is a relatively new and promising area of study in the field of regenerative medicine. The submitted article discusses the functional properties of stem cells and their potential applications in various fields, including dentistry, neurology, and oncology. The article also addresses the changes in the composition of free fatty acids during cultivation, which may have implications for the metabolic regulation of stemness and the development of stem cell-based therapies. Overall, it provides valuable information on a topic that is still being explored and has the potential to address specific gaps in the field.

Although there are other published articles on this topic, the presented article stands out by providing a comprehensive analysis of the functional properties of stem cells and the changes in free fatty acid composition during cultivation. The article provides detailed information about the experimental methods used to obtain and analyze stem cells. This may be useful for researchers in this field. Overall, this article makes a significant contribution to the relevant subject area by providing new information that sets it apart from other articles. It also offers a detailed analysis of stem cells obtained from human exfoliated deciduous teeth.

Based on the information provided in the article, the authors utilized various experimental methods to acquire and analyze the stem cells. These methods included cell culture, flow cytometry, gas chromatography coupled with mass spectrometry, and immunofluorescence staining. The file also mentions that the study was conducted in accordance with the guidelines of the Declaration of Helsinki, and informed consent was obtained from all subjects involved in the study.

The conclusions drawn from the evidence and arguments presented in the article suggest that stem cells derived from human exfoliated deciduous teeth have significant potential for use in various fields, including dentistry, neurology, and oncology. The article also suggests that the changes in the composition of free fatty acids during cultivation may have implications for the metabolic regulation of stemness and the development of stem cell-based therapies. Overall, the conclusions appear to be well-supported by the evidence and arguments presented, and they address the main question posed in the submission.

The tables and figures presented are sufficient to support the research results.

When conducting a general evaluation, it is important to clearly define the purpose and target of the study. This information should be provided in the summary and introduction sections. After making this correction, I deemed the article worthy of publication.

Author Response

Ref. Manuscript ID: ijms-2729559

Answers to Reviewer 4

Reviewer 5 Report

Comments and Suggestions for Authors

The topic of SHEDs covered in this article is very interesting and gives new perspectives on regenerative medicine and cell therapy development. The article is well-written, however, I have several suggestions below:

Please, unify abbreviations – once you use MSCs for mesenchymal stem cells, sometimes mesenchymal stem/stromal cells – it is not the same. I suggest using MSCs for mesenchymal stem/stromal cells. Once you explain the abbreviation, do not use the full name in the next sentences, like you do in the Introduction part.

In the Introduction, you wrote: “Cell plasticity and differentiation capacity towards osteogenic, chondrogenic, adipogenic, and neuronal lineage were evaluated.” Does that evaluation refer only to differentiation into neurons, or other neural cells too?

In 2.2. Multiparametric immunophenotyping analysis of SHEDs you wrote: “While no unique MSC markers have been identified so far, we observed that the proliferating cells expressed the typical MSC markers CD90, CD29, CD44, CD56, CD73, CD105, and CD117, but not the hematopoietic stem cell-related molecules CD34 and CD45.”  As far as I know, these criteria are typical for the induced MSCs. I suggest to avoid using the word “typical”. I would like to ask you to add the reference for this sentence.

“The role of CD117 in SHEDs is currently unknown, but its presence may be linked to a higher capacity for proliferation and differentiation.” Is this sentence refers to you observations? Please, specify.

Regarding all the figures, I would like you to correct the scale bars in the pictures, they are not visible. In descriptions, explain abbreviations P3, P7, P10. Change tubulin to β‐Tubulin III.

In Figure 6,  are the scale bars the same?  Cells form P3 seem to have bigger nuclei. Please, add the scale bar in the description.

Materials and methods: Have you considered adding experimental design scheme?

Author Response

Ref. Manuscript ID: ijms-2729559

Answers to Reviewer 5

Round 2

Reviewer 1 Report

Comments and Suggestions for Authors

All of the concerns have been addressed in the revised version of the manuscript. I have no further suggestions.

Author Response

Thank you for your valuable input.

Reviewer 2 Report

Comments and Suggestions for Authors

The manuscript is ready for publication 

Comments on the Quality of English Language

The manuscript is ready for publication after text editing

Author Response

Thank you for your valuable input. 

The manuscript underwent extensive language revisions to improve overall clarity.